# SMiRL: Surprise Minimizing Reinforcement Learning in Unstable Environments

**Glen Berseth**
UC Berkeley

**Daniel Geng**
UC Berkeley

**Coline Devin**
UC Berkeley

**Nicholas Rhinehart**
UC Berkeley

**Chelsea Finn**
Stanford

**Dinesh Jayaraman**
University of Pennsylvania

**Sergey Levine**
UC Berkeley

## Abstract

Every living organism struggles against disruptive environmental forces to carve out and maintain an orderly niche. We propose that such a struggle to achieve and preserve order might offer a principle for the emergence of useful behaviors in artificial agents. We formalize this idea into an unsupervised reinforcement learning method called surprise minimizing reinforcement learning (SMiRL). SMiRL alternates between learning a density model to evaluate the surprise of a stimulus, and improving the policy to seek more predictable stimuli. The policy seeks out stable and repeatable situations that counteract the environment's prevailing sources of entropy. This might include avoiding other hostile agents, or finding a stable, balanced pose for a bipedal robot in the face of disturbance forces. We demonstrate that our surprise minimizing agents can successfully play Tetris, Doom, control a humanoid to avoid falls, and navigate to escape enemies in a maze without any task-specific reward supervision. We further show that SMiRL can be used together with standard task rewards to accelerate reward-driven learning.

## 1 Introduction

Organisms can carve out environmental niches within which they can maintain relative predictability amidst the entropy around them (Boltzmann, 1886; Schrödinger, 1944; Schneider & Kay, 1994; Friston, 2009). For example, humans go to great lengths to shield themselves from surprise — we band together to build cities with homes, supplying water, food, gas, and electricity to control the deterioration of our bodies and living spaces amidst heat, cold, wind and storm. These activities exercise sophisticated control over the environment, which makes the environment more predictable and less "surprising" (Friston, 2009; Friston et al., 2009). Could the motive of preserving order guide the automatic acquisition of useful behaviors in artificial agents?

We study this question in the context of unsupervised reinforcement learning, which deals with the problem of acquiring complex behaviors and skills with no supervision (labels) or incentives (external rewards). Many previously proposed unsupervised reinforcement learning methods focus on novelty-seeking behaviors (Schmidhuber, 1991; Lehman & Stanley, 2011; Still & Precup, 2012; Bellemare et al., 2016; Houthooft et al., 2016; Pathak et al., 2017). Such methods can lead to meaningful behavior in simulated environments, such as video games, where interesting and novel events mainly happen when the agent executes a specific and coherent pattern of behavior. However, we posit that in more realistic open-world environments, natural forces outside of the agent's control *already* offer an excellent source of novelty: from other agents to unexpected natural forces, agents in these settings must contend with a constant stream of unexpected events. In such settings, rejecting perturbations and maintaining a steady equilibrium may pose a greater challenge than novelty seeking. Based on this observation, we devise an algorithm, surprise minimizing reinforcement learning (SMiRL), that specifically aims to *reduce* the entropy of the states visited by the agent.

SMiRL maintains an estimate of the distribution of visited states, $p_\theta(\mathbf{s})$, and a policy that seeks to reach likely future states under $p_\theta(\mathbf{s})$. After each action, $p_\theta(\mathbf{s})$ is updated with the new state, while the policy is conditioned on the parameters of this distribution to construct a stationary MDP. We illustrate this with a diagram in Figure 1a. We empirically evaluate SMiRL in a range of domains that

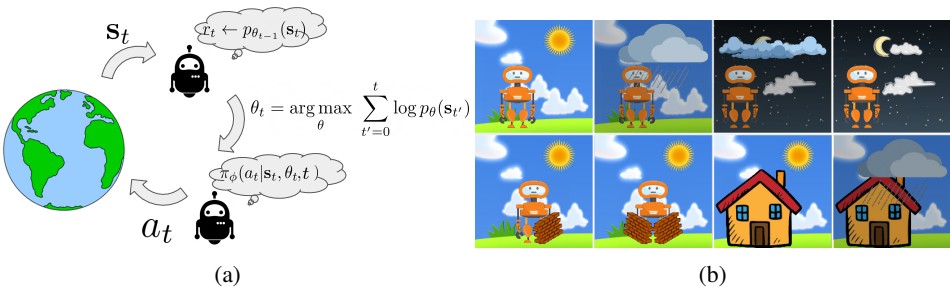

(a)                                                    (b)

Figure 1: **Left:** SMiRL observes a state $\mathbf{s}_t$ and computes a reward $r_t$ as the negative *surprise* under its current model $p_{\theta_{t-1}}(\mathbf{s}_t)$, given by $\log p_{\theta_{t-1}}(\mathbf{s}_t)$. Then the model is updated on the agents state history, including $\mathbf{s}_t$, to yield $p_{\theta_t}$. The policy $\pi_\phi(a_t|\mathbf{s}_t, \theta_t, t)$ then generates the action $a_t$. **Right:** This procedure leads to complex behavior in environments where surprising events happen on their own. In this cartoon, the robot experiences a wide variety of weather conditions when standing outside, but can avoid these surprising conditions by building a shelter, where it can reach a stable and predictable states in the long run.

are characterized by naturally increasing entropy, including video game environments based on Tetris and Doom, and simulated robot tasks that require controlling a humanoid robot to balance and walk. Our experiments show that, in environments that satisfy the assumptions of our method, SMiRL automatically discovers complex and coordinated behaviors without any reward signal, learning to successfully play Tetris, shoot enemies in Doom, and balance a humanoid robot at the edge of a cliff. We also show that SMiRL can provide an effective auxiliary objective when a reward signal is provided, accelerating learning in these domains substantially more effectively than pure novelty-seeking methods. Videos of our results are available online[1]

## 2 RELATED WORK

Prior work on unsupervised learning has proposed algorithms that learn without a reward function, such as empowerment (Klyubin et al., 2005; Mohamed & Jimenez Rezende, 2015) or intrinsic motivation (Chentanez et al., 2005; Oudeyer & Kaplan, 2009; Oudeyer et al., 2007). Intrinsic motivation has typically focused on encouraging novelty-seeking behaviors by maximizing model uncertainty (Houthooft et al., 2016; Still & Precup, 2012; Shyam et al., 2018; Pathak et al., 2019), by maximizing model prediction error or improvement (Lopes et al., 2012; Pathak et al., 2017), through state visitation counts (Bellemare et al., 2016), via surprise maximization (Achiam & Sastry, 2017b; Schmidhuber, 1991; Sun et al., 2011), and through other novelty-based reward bonuses (Lehman & Stanley, 2011; Achiam & Sastry, 2017a; Burda et al., 2018a; Kim et al., 2019). We do the opposite. Inspired by the free energy principle (Friston, 2009; Friston et al., 2009; Ueltzhöffer, 2018; Faraji et al., 2018; Friston et al., 2016) including recent methods that train policies using RL (Tschantz et al., 2020a;b; Annabi et al., 2020) that encode a prior over desired observations, we instead incentivize an agent to *minimize* surprise over the distribution of states generated by the policy in unstable environments, and study the resulting behaviors. In such environments it is non-trivial to achieve low entropy state distributions, which we believe are more reflective of the real world. Learning progress methods that minimize model parameter entropy (Lopes et al., 2012; Kim et al., 2020) avoid the issues novelty-based methods have with noisy distractors. These methods are based on learning the parameters of the dynamics where our method is learning to control the marginal state distribution.

Several works aim to maximize state entropy to encourage exploration (Lee et al., 2019; Hazan et al., 2019). Our method aims to do the opposite, *minimizing* state entropy. Recent work connects the free energy principle, empowerment and predictive information maximization under the same framework to understand their differences (Biehl et al., 2018). Existing work has also studied how competitive self-play and competitive, multi-agent environments can lead to complex behaviors with minimal reward information (Silver et al., 2017; Bansal et al., 2017; Sukhbaatar et al., 2017; Baker et al., 2019; Weihs et al., 2019; Chen et al., 2020). Like these works, we also consider how complex behaviors can emerge in resource-constrained environments, but instead of multi-agent competition, we utilize surprise minimization to drive the emergence of complex skills.

---

[1]https://sites.google.com/view/surpriseminimization

## 3   SURPRISE MINIMIZING AGENTS

We propose surprise minimization as a means to operationalize the idea of learning useful behaviors by seeking out low entropy state distributions. The long term effects of actions on surprise can be subtle, since actions change both (i) the state that the agent is in, and (ii) its beliefs, represented by a model $p_\theta(\mathbf{s})$, about which states are likely under its current policy. SMiRL induces the agent to modify its policy $\pi$ so that it encounters states $\mathbf{s}$ with high $p_\theta(\mathbf{s})$, as well as to seek out states that will change the model $p_\theta(\mathbf{s})$ so that future states are more likely. In this section, we will first describe what we mean by unstable environments and provide the surprise minimization problem statement, and then present our practical deep reinforcement learning algorithm for learning policies that minimize surprise.

Many commonly used reinforcement learning benchmark environments are *stable*, in the sense the agent remains in a narrow range of starting states unless it takes coordinated and purposeful actions. In such settings, unsupervised RL algorithms that seek out novelty can discover meaningful behaviors. However, many environments – including, as we argue, those that reflect properties commonly found in the real world, – are unstable, in the sense that unexpected and disruptive events naturally lead to novelty and increased state entropy even if the agent does not carry out any particularly meaningful or purposeful behavior. In unstable environments, minimizing cumulative surprise requires taking actions to reach a stable distribution of states, and then acting continually and purposefully to stay in this distribution. An example of this is illustrated in Figure 1b: the agent's environment is unstable due to varied weather. If the robot builds a shelter, it will initially experience unfamiliar states, but in the long term the observations inside the shelter are more stable and less surprising than those outside. Another example is the game of Tetris (Figure 2), where the environment spawns new blocks and drops them into random configurations, unless a skilled agent takes actions to control the board. The challenge of maintaining low entropy in unstable settings forces the SMiRL agent to acquire meaningful skills. We defer a more precise definition of unstable environments to Section 4, where we describe several unstable environments and contrast them with the static environments that are more commonly found in RL benchmark tasks. In static environments, novelty seeking methods must discover complex behaviors to increase entropy, leading to interesting behavior, while SMiRL may trivially find low entropy policies. We show that the reverse is true for unstable environments: a novelty seeking agent is satisfied with watching the environment change around it, while a surprise minimizing agent must develop meaningful skills to lower entropy.

**Problem statement.**   To instantiate SMiRL, we design a reinforcement learning agent that receives larger rewards for experiencing more familiar states, based on the history of states it has experienced during the current episode. This translates to learning a policy with the lowest state entropy. We assume a fully-observed controlled Markov process (CMP), where we use $\mathbf{s}_t$ to denote the state at time $t$, $a_t$ to denote the agent's action, $p(\mathbf{s}_0)$ to denote the initial state distribution, and $T(\mathbf{s}_{t+1}|\mathbf{s}_t, a_t)$ to denote the transition probabilities. The agent learns a policy $\pi_\phi(a|\mathbf{s})$, parameterized by $\phi$. The goal is to minimize the entropy of its state marginal distribution under its current policy $\pi_\phi$ at each time step of the episode. We can estimate this entropy by fitting an estimate of the state marginal $d^{\pi_\phi}(\mathbf{s}_t)$ at each time step $t$, given by $p_{\theta_{t-1}}(\mathbf{s}_t)$, using the states seen so far during the episode, $\tau_t = \{\mathbf{s}_1, \ldots, \mathbf{s}_t\}$ that is stationary. The sum of the entropies of the state distributions over an episode can then be estimated as

$$\sum_{t=0}^{T} \mathcal{H}(\mathbf{s}_t) = -\sum_{t=0}^{T} \mathbb{E}_{\mathbf{s}_t \sim d^{\pi_\phi}(\mathbf{s}_t)}[\log d^{\pi_\phi}(\mathbf{s}_t)] \leq -\sum_{t=0}^{T} \mathbb{E}_{\mathbf{s}_t \sim d^{\pi_\phi}(\mathbf{s}_t)}[\log p_{\theta_{t-1}}(\mathbf{s}_t)], \qquad (1)$$

where the inequality becomes an equality if $p_{\theta_{t-1}}(\mathbf{s}_t)$ accurately models $d^{\pi_\phi}(\mathbf{s}_t)$. Minimizing the right-hand side of this equation corresponds to maximizing an RL objective with rewards:

$$r(\mathbf{s}_t) = \log p_{\theta_{t-1}}(\mathbf{s}_t). \qquad (2)$$

However, an optimal policy for solving this problem must take changes in the distribution $p_{\theta_{t-1}}(\mathbf{s}_t)$ into account when selecting actions, since this distribution changes at each step. To ensure that the underlying RL optimization corresponds to a stationary and Markovian problem, we construct an *augmented* MDP to instantiate SMiRL in practice, which we describe in the following section.

**Training SMiRL agents.** In order to instantiate SMiRL, we construct an *augmented* MDP out of the original CMP, where the reward in Equation (2) can be expressed entirely as a function of the state. This augmented MDP has a state space that includes the original state $\mathbf{s}_t$, as well as the sufficient statistics of $p_{\theta_t}(\mathbf{s})$. For example, if $p_{\theta_t}(\mathbf{s})$ is a normal distribution with parameters $\theta_t$, then $(\theta_t, t)$ – the parameters of the distribution and the number of states seen so far – represents a sufficient statistic. Note that it is possible to use other, more complicated, methods to summarize the statistics, including reading in the entirety of $\tau_t$ using a recurrent model. The policy conditioned on the augmented state is then given by $\pi_\phi(a_t|\mathbf{s}_t, \theta_t, t)$. The parameters of the sufficient statistics are updated $\theta_t = \mathcal{U}(\tau_t)$ using a maximum likelihood state density estimation process

---

**Algorithm 1** SMiRL

1: **while** not converged **do**
2:     $\beta \leftarrow \{\}$                            ▷ Reset experience
3:     **for** episode $= 0, \ldots, M$ **do**
4:         $\mathbf{s}_0 \sim p(\mathbf{s}_0); \tau_0 \leftarrow \{\mathbf{s}_0\}$     ▷ Initialize state
5:         $\bar{\mathbf{s}}_0 \leftarrow (\mathbf{s}_0, \mathbf{0}, 0)$         ▷ Initialize aug. state
6:         **for each** $t = 0, \ldots, T$ **do**
7:             $a_t \sim \pi_\phi(a_t|\mathbf{s}_t, \theta_t, t)$        ▷ Get action
8:             $\mathbf{s}_{t+1} \sim T(\mathbf{s}_{t+1}|\mathbf{s}_t, a_t)$ ▷ Step dynamics
9:             $r_t \leftarrow \log p_{\theta_t}(\mathbf{s}_{t+1})$     ▷ SMiRL reward
10:            $\tau_{t+1} \leftarrow \tau_t \cup \{\mathbf{s}_{t+1}\}$        ▷ Record state
11:            $\theta_{t+1} \leftarrow \mathcal{U}(\tau_{t+1})$               ▷ Fit model
12:            $\bar{\mathbf{s}}_{t+1} \leftarrow \{(\mathbf{s}_{t+1}, \theta_{t+1}, t_{t+1})\}$
13:            $\beta \leftarrow \beta \cup \{(\bar{\mathbf{s}}_t, a_t, r_t, \bar{\mathbf{s}}_{t+1})\}$
14:        **end for**
15:    **end for each**
16:    $\phi \leftarrow \texttt{RL}(\phi, \beta)$                    ▷ Update policy
17: **end while**

---

$\theta_t = \arg\max_\theta \sum_{n=0}^{t} \log p_\theta(\mathbf{s}_n)$ over the experience within the episode $\tau_t$. When $(\theta_t, t)$ is a sufficient statistic, the update may be written as $\theta_t = \mathcal{U}(\mathbf{s}_t, \theta_{t-1}, t-1)$. Specific update functions $\mathcal{U}(\tau_t)$ used in our experiments are described in Appendix C and at the end of the section. Since the reward is given by $r(\mathbf{s}_t, \theta_{t-1}, t-1) = \log p_{\theta_{t-1}}(\mathbf{s}_t)$, and $\theta_t$ is a function of $\mathbf{s}_t$ and $(\theta_{t-1}, t-1)$, the resulting RL problem is fully Markovian and stationary, and as a result standard RL algorithms will converge to locally optimal solutions. Appendix D include details on the MDP dynamics. In Figure 8, we illustrate the evolution of $p_{\theta_t}(\mathbf{s})$ during an episode of the game Tetris. The pseudocode for this algorithm is presented in Algorithm 1.

**Density estimation with learned representations.** SMiRL may, in principle, be used with any choice of model class for the density model $p_{\theta_t}(\mathbf{s})$. As we show in our experiments, relatively simple distribution classes, such as products of independent marginals, suffice to run SMiRL in many environments. However, it may be desirable in more complex environments to use more sophisticated density estimators, especially when learning directly from high-dimensional observations such as images. In these cases, we can use variational autoencoders (VAEs) (Kingma & Welling, 2014) to learn a non-linear state representation. A VAE is trained using the standard ELBO objective to reconstruct states $\mathbf{s}$ after encoding them into a latent representation $\mathbf{z}$ via an encoder $q_\omega(\mathbf{z}|\mathbf{s})$, with parameters $\omega$. Thus, $\mathbf{z}$ can be viewed as a compressed representation of the state.

When using VAE representations, we train the VAE online together with the policy. This approach necessitates two changes to the procedure described Algorithm 1. First, training a VAE requires more data than the simpler independent models, which can easily be fitted to data from individual episodes. We propose to overcome this by not resetting the VAE parameters between training episodes, and instead training the VAE across episodes. Second, instead of passing the VAE model parameters to the policy, we only update a distribution over the VAE latent state, given by $p_{\theta_t}(\mathbf{z})$, such that $p_{\theta_t}(\mathbf{z})$ replaces $p_{\theta_t}(\mathbf{s})$ in the SMiRL algorithm, and is fitted to only that episode's (encoded) state history. We represent $p_{\theta_t}(\mathbf{z})$ as a normal distribution with a diagonal covariance, and fit it to the VAE encoder outputs. Thus, the mean and variance of $p_{\theta_t}(\mathbf{z})$ are passed to the policy at each time step, along with $t$. This implements the density estimate in line 9 of Algorithm 1. The corresponding update $\mathcal{U}(\tau_t)$ is:

$$\mathbf{z}_0, \ldots, \mathbf{z}_t = \mathbb{E}[q_\omega(\mathbf{z}|\mathbf{s})] \text{ for } \mathbf{s} \in \tau_t, \quad \mu = \frac{1}{t+1}\sum_{j=0}^{t} \mathbf{z}_j, \sigma = \frac{1}{t+1}\sum_{j=0}^{t}(\mu - \mathbf{z}_j)^2, \theta_t = [\mu, \sigma].$$

Training the VAE online, over all previously seen data, deviates from the recipe in the previous section, where the density model was only updated *within* an episode. In this case the model is updated after a collection of episodes. This makes the objective for RL somewhat non-stationary and could theoretically cause issues for convergence, however we found in practice that the increased representational capacity provides significant improvement in performance.

## 4 EVALUATION ENVIRONMENTS

We evaluate SMiRL on a range of environments, from video game domains to simulated robotic control scenarios. In these unstable environments, the world evolves automatically, without the goal-driven behavior of the agent, due to disruptive forces and adversaries. Standard RL benchmark tasks are typically static, in the sense that unexpected events don not happen unless the agent carries out a specific and coordinated sequence of actions. We therefore selected these environments specifically to be *unstable*, as we discuss below. This section describes each environment, with details of the corresponding MDPs in Appendix B. Illustrations of the environments are shown in Figure 2.

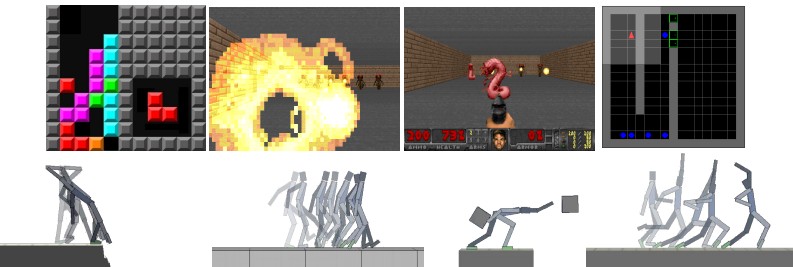

Figure 2: Evaluation environments. Top row, left to right: *Tetris* environment, *VizDoom TakeCover* and *DefendTheLine*, *HauntedHouse* with pursuing "enemies," where the agent can reach a more stable state by finding the doors and leaving the region with enemies. Bottom row, left to right: *Humanoid* next to a *Cliff*, *Humanoid* on a *Treadmill*, *Pedestal*, *Humanoid* learning to walk.

***Tetris.*** The classic game offers a naturally unstable environment — the world evolves according to its own dynamics even in the absence of coordinated agent actions, piling pieces and filling the board. The agent's task is to place randomly supplied blocks to construct and eliminate complete rows. The environment gives a reward of $-1$ when the agent fails or dies by stacking a column too high. Otherwise, the agent gets $0$.

***VizDoom.*** We consider two *VizDoom* environments from Kempka et al. (2016): *TakeCover* and *DefendTheLine* where enemies throw fireballs at the agent, which can move around to avoid damage. *TakeCover* is unstable and evolving, with new enemies appearing over time and firing at the player. The agent is evaluated on how many fireballs hit it, which we term the "damage" taken by the agent.

***HauntedHouse.*** This is a partially observed navigation task. The agent (red) starts on the left of the map, and is pursued by "enemies" (blue). To escape, the agent can navigate down the hallways and through randomly placed doors (green) to reach the *safe* room on the right, which the enemies cannot enter. To get to the *safe* room the agent must endure increased surprise early on, since the doors appear in different locations in each episode.

**Simulated *Humanoid* robots.** A simulated planar *Humanoid* agent must avoid falling in the face of external disturbances (Berseth et al., 2018). We evaluate four versions of this task. For *Cliff* the agent is initialized sliding towards a cliff, for *Treadmill*, the agent is on a small platform moving backwards at $1$ m/s. In *Pedestal*, random forces are applied to it, and objects are thrown at it. In *Walk*, we evaluate how the SMiRL reward stabilizes an agent that is learning to walk. In all four tasks, we evaluate the proportion of episodes the robot does not fall.

**Training Details.** For discrete action environments, the RL algorithm used is DQN (Mnih et al., 2013) with a target network. For the *Humanoid* domains, we use TRPO (Schulman et al., 2015). For *Tetris* and the *Humanoid* domains, the policies are parameterized by fully connected neural networks, while *VizDoom* uses a convolutional network. Additional details are in Appendix Section B.

**Environment Stability.** In Section 3, we described the connection between SMiRL and unstable environments. We can quantify how *unstable* an environment is by computing a *relative entropy gap*. We compare the entropy between three methods: entropy minimizing (SMiRL), entropy maximizing (RND) methods, and an initial random (Random) policy (or, more generally, an uninformed policy, such as a randomly initialized neural network). In stable environments, an uninformed random policy would only attain slightly higher state entropy than one that minimizes the entropy explicitly (SMiRL - Random $\sim 0$), whereas a novelty-seeking policy should attain much higher entropy (RND - Random $> 0$), indicating a relative entropy gap in the *positive* direction. In an unstable environment, we

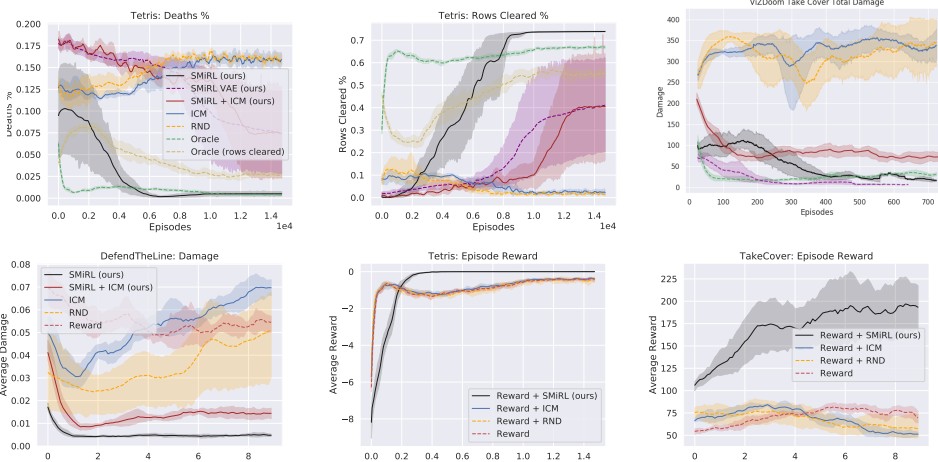

Figure 3: Comparison between SMiRL, ICM, RND, and an Oracle baseline that uses the true reward, evaluated on *Tetris* with (top-left) number of deaths per episode (lower is better), (top-center) rows cleared per episode (higher is better), and in *TakeCover* (top-right) and *DefendTheLine* (bottom-left) on amount of damage taken (lower is better). In all cases, the RL algorithm used for training is DQN, and all results are averaged over 6 random seeds, with the shaded areas indicating the standard deviation. In *Tetris* (bottom-center) and *TakeCover* (bottom-right) methods are evaluated on how they improve learning when added to the environment reward function.

expect random policies and novelty-seeking policies should attain similar entropies, whereas entropy minimization should result in much lower entropy (SMiRL - Rand < 0), indicating a *negative* entropy gap. To compute the entropy used in this evaluation, we used the approximation in Eq. 1 multiplied by $-1$ for three of our tasks as well as many Atari games studied in the RND paper (Burda et al., 2018b), with numbers shown in Table 1 and full results in Appendix E. Our environments have a large *negative* entropy gap, whereas most Atari games lack this clear entropy gap.[2] We therefore expect SMiRL to perform well on these tasks, which we use in the next section, but poorly on most Atari games. We show animations of the resulting policies on our anonymous project website.

## 5 EXPERIMENTAL RESULTS

Our experiments aim to answer the following questions: **(1)** Can SMiRL learn meaningful and complex emergent behaviors without supervision? **(2)** Can we improve SMiRL by incorporating representation learning via VAEs, as described in Section 3? **(3)** Can SMiRL serve as a joint training objective to accelerate the acquisition of reward-guided behavior, and does it outperform prior intrinsic motivation methods in this role? We also illustrate several applications of SMiRL, showing that it can accelerate task learning, facilitate exploration, and implement a form of imitation learning. Video results of learned behaviors are available at https://sites.google.com/view/surpriseminimization

| Environment | RND* | SMiRL* | Relative |
|---|---|---|---|
| *DefendTheLine* | -0.3±0.6 | -43.1±0.4 | -43.4 |
| *Tetris* | 1.5±2.7 | -11.9±2.1 | -10.4 |
| *TakeCover* | -1.2±0.7 | -7.3±0.7 | -8.5 |
| *Assault* | 11.3±1.4 | -56.9±2.3 | -45.6 |
| *SpaceInvaders* | 1.9±3.4 | -10.2±4.2 | -8.3 |
| *Carnival* | 20.4±1.4 | -23.1±4.3 | -2.7 |
| *RiverRaid* | -5.5±3.4 | 5.8±3.2 | 0.3 |
| *Gravitar* | 30.8±1.7 | -26.5±1.3 | 4.3 |
| *Berzerk* | 17.2±1.4 | -2.9±4.7 | 14.3 |

Table 1: Difference in entropy vs. a *Random* policy (SMiRL*=SMiRL-Random and RND*=RND-Random, Relative=RND*+SMiRL*). More negative values indicate more unstable environments. Note the *negative* relative entropy gap on our tasks and for *Assault* and *SpaceInvaders*.

## 5.1 Emergent Behavior with Unsupervised Learning

To answer **(1)**, we evaluate SMiRL on the *Tetris*, *VizDoom* and *Humanoid* tasks, studying its ability to generate purposeful coordinated behaviors without engineered task-specific rewards. We compare SMiRL to two intrinsic motivation methods, ICM (Pathak et al., 2017) and RND (Burda et al., 2018b), which seek out states that *maximize* surprise or novelty. For reference, we also include an Oracle baseline that directly optimizes the task reward. We find that SMiRL acquires meaningful emergent behaviors across these domains. In both the *Tetris* and *VizDoom* environments, stochastic and chaotic events force the SMiRL agent to take a coordinated course of action to avoid unusual states, such as full Tetris boards or fireball explosions. On *Tetris*, after training for 3000 epochs, SMiRL achieves near-perfect play, on par with the oracle baseline, with no deaths, indicating that SMiRL may provide better dense rewards than the *Oracle* reward, as shown in Figure 3 (top-left, top-middle). Figure 3 top-left and top-center show data from the same experiment that plots two different metrics, where the *Oracle* is optimized for minimizing deaths. We include another oracle, *Oracle (rows cleared)* where the reward function is the number of rows cleared. ICM and RND seek novelty by creating more and more distinct patterns of blocks rather than clearing them, leading to deteriorating game scores over time. The SMiRL agent also learns emergent game playing behavior in *VizDoom*, acquiring an effective policy for dodging the fireballs thrown by the enemies, illustrated in Figure 3 (top-right and bottom-left). Novelty-seeking seeking methods once again yield deteriorating rewards over time. In *Cliff*, the SMiRL agent learns to brace against the ground and stabilize itself at the edge, as shown in Figure 2. In *Treadmill*, SMiRL learns to jump forward to increase the time it stays on the treadmill. In *Pedestal*, the agent must actively respond to persistent disturbances. We find that SMiRL learns a policy that can reliably keep the agent atop the pedestal, as shown in Figure 2. Figure 4 plots the reduction in falls in the *Humanoid* environments. Novelty-seeking methods learn irregular behaviors that cause the humanoid to jump off the *Cliff* and *Pedestal* tasks and roll around on the *Treadmill*, maximizing the variety (and quantity) of falls.

Next, we study how representation learning with a VAE improves the SMiRL algorithm (question **(2)**). In these experiments, we train a VAE model and estimate surprise in the VAE latent space. This leads to faster acquisition of the emergent behaviors for *TakeCover* (Figure 3, top-right), *Cliff* (Figure 4, left), and *Treadmill* (Figure 4, middle), where it also leads to a more successful locomotion behavior.

At first glance, the SMiRL surprise minimization objective appears to be the opposite of standard intrinsic motivation objectives (Bellemare et al., 2016; Pathak et al., 2017; Burda et al., 2018b) that seek out states with *maximal* surprise (i.e., novel states). However, while those approaches measure surprise with respect to all prior experience, SMiRL minimizes surprise over each episode. We demonstrate that these two approaches are in fact complementary. SMiRL can use conventional intrinsic motivation methods to aid in exploration *so as to discover more effective policies for minimizing surprise*. We can, therefore, combine these two methods and learn more sophisticated behaviors. While SMiRL on its own does not successfully produce a good walking gait on *Treadmill*, the addition of novelty-seeking intrinsic motivation allows increased exploration, which results in an improved walking gait that remains on the treadmill longer, as shown in Figure 4 (middle). We

---

[2]We expect that in all cases, random policies will have somewhat higher state entropy than SMiRL, so the entropy gap should be interpreted in a relative sense.

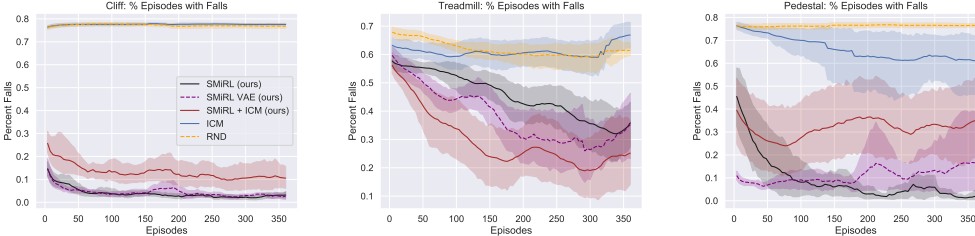

Figure 4: *Cliff*, *Treadmill* and *Pedestal* results. In all cases, SMiRL reduces episodes with falls (lower is better). SMiRL that uses the VAE for representation learning typically attains better performance. Trained using TRPO with results averaged over 12 random seeds, showing mean and standard deviation in the shaded area.

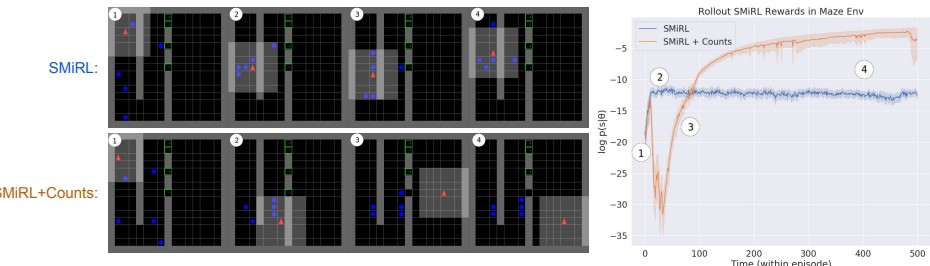

Figure 5: Here we show SMiRL's incentive for longer-term planning in the *HauntedHouse* environment. On the top-left, we see that SMiRL on its own does not explore well enough to reach the *safe* room on the right. Adding exploration via *Counts* (bottom-left) allows SMiRL to discover more optimal entropy reducing policies, shown on the right.

evaluate this combined approach across environments including *Pedestal* and *Cliff* as well, where learning to avoid falls is also a challenge. For these two tasks SMiRL can already discover strong surprise minimizing policies and adding exploration bonuses does not provide additional benefit. In Figure 5 adding a bonus enables the agent to discover improved surprise minimizing strategies.

**SMiRL and long term surprise.** Although the SMiRL objective by itself does not specifically encourage exploration, we observe that optimal SMiRL policies exhibit active "searching" behaviors, seeking out objects in the environment that would allow for reduced long-term surprise. For example, in *HauntedHouse*, the positions of the doors leading to the safe room change between episodes, and the policy trained with SMiRL learns to search for the doors to facilitate lower future surprise, even if finding the doors themselves yields higher short-term surprise. This behavior is illustrated in Figure 5, along with the "delayed gratification" plot, which shows that the SMiRL agent incurs higher surprise early in the episode, for the sake of much lower surprise later.

## 5.2 APPLICATIONS OF SMIRL

While the focus of this paper is on the emergent behaviors obtained by SMiRL, here we study more pragmatic applications. We show that SMiRL can be used for basic imitation and joint training to accelerate reward-driven learning.

**Imitation.** SMiRL can be adapted to perform imitation by initializing the prior via the buffer $\mathcal{D}_0$ with states from demonstrations, or individual desired outcome states. We initialize the buffer $\mathcal{D}_0$ in *Tetris* with user-specified *desired* board states. An illustration of the *Tetris* imitation task is presented in Figure 6, showing imitation of a box pattern (top) and a checkerboard pattern (bottom), with the leftmost frame showing the user-specified example, and the other



Figure 6: Tetris *imitation* by starting $p_\theta(\mathbf{s})$ with left image.

frames showing actual states reached by the SMiRL agent. While several prior works have studied imitation without example actions (Liu et al., 2018; Torabi et al., 2018a; Aytar et al., 2018; Torabi et al., 2018b; Edwards et al., 2018; Lee et al.), this capability emerges automatically in SMiRL, without any further modification to the algorithm.

**SMiRL as an auxiliary reward.** We explore how combining SMiRL with a task reward can lead to faster learning. We hypothesize that, when the task reward is aligned with avoiding unpredictable situations (e.g., falling or dying), adding SMiRL as an auxiliary reward can accelerate learning by providing a dense intermediate signal. The full reward is given by $r_{\text{combined}}(\mathbf{s}) = r_{\text{task}}(\mathbf{s}) + \alpha r_{\text{SMiRL}}(\mathbf{s})$, where $\alpha$ is chosen to put the two reward terms at a similar magnitude. We study this application of SMiRL in the tasks: *Tetris* in Figure 3 (bottom-center), *TakeCover* in Figure 3 (bottom-right), *DefendTheLine* and *Walk*. On the easier tasks *Tetris* and *TakeCover* task (Figure 7), prior exploration methods generally lead to significantly worse performance and SMiRL improves learning speed. On the harder *Walk* and *DefendTheLine* tasks, the SMiRL reward accelerates learning substantially, and also significantly reduces the number of falls or deaths. We found that increasing the difficulty of *TakeCover* and *DefendTheLine* (via the environment's difficulty setting (Kempka et al., 2016)) resulted in a clearer separation between SMiRL and other methods

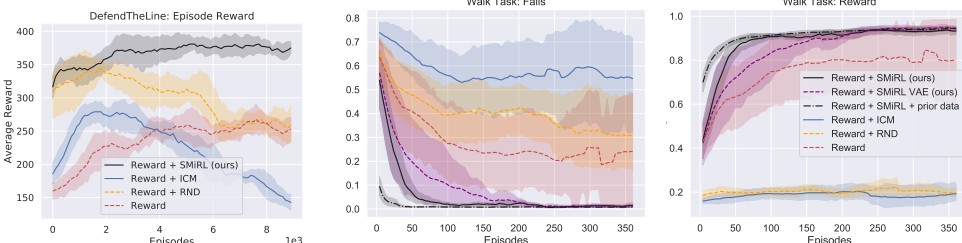

Figure 7: Left: We combine SMiRL with the survival time task reward in the *DefendTheLine* task. Middle/Right: We combine the SMiRL reward with the *Walk* reward and initialize SMiRL without walking prior walking data (ours) and with (prior data). Results over 12 seeds with standard deviation indicated by the shaded area.

In *Walk*, we include a version of SMiRL with *prior data*, where $p_\theta(\mathbf{s})$ is initialized with 8 walking trajectories (256 timesteps each), similar to the imitation setting. Incorporating prior data requires no modification to the SMiRL algorithm, and we can see in Figure 7 (middle and right) that this variant ("Reward + SMiRL + prior data") further accelerates learning and reduces the number of falls. This shows that while SMiRL can learn from scratch, it is possible to encode prior knowledge in $p_\theta(\mathbf{s})$ to improve learning.

## 6 DISCUSSION

We presented an unsupervised reinforcement learning method based on *minimizing* surprise. We show that surprise minimization can be used to learn a variety of behaviors that reach "homeostasis," putting the agent into stable state distributions in its environment. Across a range of tasks, these cycles correspond to useful, semantically meaningful, and complex behaviors: clearing rows in *Tetris*, avoiding fireballs in *VizDoom*, and learning to balance and hop with a bipedal robot. The key insight utilized by our method is that, in contrast to simple simulated domains, realistic environments exhibit unstable phenomena that gradually increase entropy over time. An agent that resists this growth in entropy must take effective and coordinated actions, thus learning increasingly complex behaviors. This stands in contrast to commonly proposed intrinsic exploration methods based on novelty.

Besides fully unsupervised reinforcement learning, where we show that our method can give rise to intelligent and sophisticated policies, we also illustrate several more practical applications of our approach. We show that surprise minimization can provide a general-purpose auxiliary reward that, when combined with task rewards, can improve learning in environments where avoiding catastrophic (and surprising) outcomes is desirable. We also show that SMiRL can be adapted to perform a rudimentary form of imitation.

Our investigation of surprise minimization suggests several directions for future work. The particular behavior of a surprise minimizing agent is strongly influenced by the choice of state representation: by including or excluding particular observation modalities, the agent will be more or less surprised. Thus, tasks may be designed by choosing an appropriate state or observation representations. Exploring this direction may lead to new ways of specifying behaviors for RL agents without explicit reward design. Other applications of surprise minimization may also be explored in future work, possibly for mitigating reward misspecification by disincentivizing any unusual behavior that likely deviates from what the reward designer intended. The experiments in this work make use of available or easy to learn state representations. Using these learned representations does not address the difficulty of estimating and minimizing surprise across episodes or more generally over long sequences (possibly a single episode) which is a challenge for surprise minimization-based methods. We believe that non-episodic surprise minimization is a promising direction for future research to study how surprise minimization can result in intelligent and sophisticated behavior that maintains homeostasis by acquiring increasingly complex behaviors.

**Acknowledgments** The authors thank Aviral Kumar and Michael Janner for discussion. This research was supported by a DARPA Young Faculty Award #D13AP0046, Office of Naval Research, the National Science Foundation, NVIDIA, Amazon, and ARL DCIST CRA W911NF-17-2-0181.

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

## A  STATE ENTROPY MINIMIZATION DERIVATION

Here we will show that the SMiRL reward function leads to a policy objective that lower-bounds the negative entropy of the state marginal distribution, $-H(d^{\pi_\phi})$. In the infinite horizon setting, the value of a trajectory $\tau = (\mathbf{s}_0, a_0, \mathbf{s}_1, a_1, \dots)$ is given as the discounted cumulative rewards: $R(\tau) = (1 - \gamma)\sum_{t=0}^{\infty}\gamma^t r(\mathbf{s}_t, a_t)$. In our case, $r(\mathbf{s}_t, a_t)$ is a function only of state: $r(\mathbf{s}_t, a_t) = r(\mathbf{s}_t) = \log p_\theta(\mathbf{s}_t)$. The policy and dynamics define a trajectory distribution $p(\tau|\phi) = p(\mathbf{s}_0)\prod_{t=1}^{\infty}p(\mathbf{s}_{t+1}|\mathbf{s}_t, a_t)\pi_\phi(a_t|\mathbf{s}_t)$. The value of a policy is its expected cumulative reward:

$$V^{\pi_\phi} = \mathbb{E}_{\tau \sim p(\tau|\pi_\phi)}R(\tau) = (1-\gamma)\mathbb{E}_{\tau \sim p(\tau|\pi_\phi)}\sum_{t=0}^{\infty}\gamma^t r(\mathbf{s}_t).$$

Using the indicator function $\mathbb{1}(a = b) \triangleq 1$ if $a = b$; $0$ if $a \neq b$, the $t$-step state distribution and the discounted state marginal are given as:

$$d_t^{\pi_\phi}(\mathbf{s}) = p(\mathbf{s}_t = \mathbf{s}|\pi_\phi) = \mathbb{E}_{\tau \sim P(\tau|\pi_\phi)}\mathbb{1}(\mathbf{s}_t = \mathbf{s})$$

$$d^{\pi_\phi}(\mathbf{s}) = (1 - \gamma)\sum_{t=0}^{\infty}\gamma^t d_t^{\pi_\phi}(\mathbf{s})$$

The expected reward under the discounted state marginal is equivalent to the policy value $V^\pi$:

$$\mathbb{E}_{s \sim d^{\pi_\phi}(\mathbf{s})}[r(\mathbf{s})] = \int d^{\pi_\phi}(\mathbf{s})r(\mathbf{s})\mathrm{d}s$$

$$= (1-\gamma)\mathbb{E}_{\tau \sim P(\tau|\pi_\phi)}\sum_{t=0}^{\infty}\gamma^t \int \mathbb{1}(\mathbf{s}_t = \mathbf{s})r(\mathbf{s})\mathrm{d}s$$

$$= (1-\gamma)\mathbb{E}_{\tau \sim P(\tau|\pi_\phi)}\sum_{t=0}^{\infty}\gamma^t r(\mathbf{s}_t) = V^{\pi_\phi}$$

After incorporating the rewards, the policy value becomes:

$$V^{\pi_\phi} = \mathbb{E}_{s \sim d^{\pi_\phi}(\mathbf{s})}[r(\mathbf{s})] = \mathbb{E}_{s \sim d^{\pi_\phi}(\mathbf{s})}[\log p_\theta(\mathbf{s})] = J(\phi, \theta)$$

$$J(\phi, \theta) = -H(d^{\pi_\phi}, p_\theta) \leq -H(d^{\pi_\phi}),$$

where $H(d^\pi, p_\theta)$ denotes the cross-entropy between $d^{\pi_\phi}$ and $p_\theta$. Thus, by optimizing $\pi_\phi$ with reward function $\log p_\theta(\mathbf{s})$ via RL, we maximize the policy value, equivalent to the negative cross-entropy from the discounted state marginal and the model. By optimizing $p_\theta$ with maximum-likelihood density estimation (minimizing forward cross-entropy) of states induced by $\pi_\phi$, we tighten the bound towards $-H(d^{\pi_\phi}(\mathbf{s}))$. When the model is perfect (i.e., $p_\theta = d^{\pi_\phi}$), the inequality becomes tight. As discussed in the main text, we cannot draw samples from $d^{\pi_\phi}(\mathbf{s})$. We can only sample trajectories of finite length $T$ by rolling out the policy $\pi_\phi$. In this case, the finite-horizon discounted state marginal can be written as:

$$\hat{d}^{\pi_\phi, T}(\mathbf{s}) \triangleq \frac{1 - \gamma}{1 - \gamma^T}\sum_{t=0}^{T-1}\gamma^t p(\mathbf{s}_t = \mathbf{s}|\pi_\phi, t < T)$$

$$= \frac{1 - \gamma}{1 - \gamma^T}\sum_{t=0}^{T-1}\gamma^t \mathbb{E}_{\tau \sim p(\tau|\pi_\phi)}\mathbb{1}(\mathbf{s}_t = \mathbf{s}, t < T).$$

Note that $d^{\pi_\phi, T}(\mathbf{s}) \geq 0 \quad \forall \mathbf{s}$, and $\sum_{\mathbf{s}} d^{\pi_\phi, T}(\mathbf{s}) = \frac{1-\gamma}{1-\gamma^T}\sum_{t=0}^{T-1}\gamma^t \sum_{\mathbf{s}} p(\mathbf{s}_t = \mathbf{s}|\pi_\phi, t < T) = 1$.

$d^{\pi_\phi, T}(\mathbf{s})$ converges to $d^{\pi_\phi}(\mathbf{s})$ as $T \to \infty$: $\lim_{T\to\infty}\hat{d}^{\pi_\phi, T} = (1-\gamma)\sum_{t=0}^{\infty}\gamma^t \mathbb{E}_{P(\tau|\pi_\phi)}\mathbb{1}(\mathbf{s}_t = \mathbf{s}) = d^{\pi_\phi}$.

Thus, by using $d^{\pi_\phi, T}(\mathbf{s})$ in place of $d^{\pi_\phi}(\mathbf{s})$, we obtain an objective, $-H(\hat{d}^{\pi_\phi, T}(\mathbf{s}), p_\theta(\mathbf{s}))$, that we can approximate with a sample of finite-length trajectories and optimize with respect to $\phi$ using a

policy-gradient reinforcement learning algorithm on the equivalent finite-horizon value function:

$$\bar{J}(\phi; \theta) = -H(\hat{d}^{\pi_\phi, T}(\mathbf{s}), p_\theta(\mathbf{s})) = V^{\pi_\phi, T}$$

$$= \frac{1 - \gamma}{1 - \gamma^T} \mathbb{E}_{\tau \sim P(\tau | \pi_\phi)} \sum_{t=0}^{T-1} \gamma^t \log p_\theta(\mathbf{s}_t).$$

The approximation to $J(\phi; \theta)$ improves as $T \to \infty$, since $\lim_{T \to \infty} \hat{d}^{\pi_\phi, T}(\mathbf{s}) = d^{\pi_\phi}$.

## B    ADDITIONAL IMPLEMENTATION DETAILS

**Additional Training Details.**    The experiments in the paper used two different RL algorithms for discrete action environemnts (Double DQN) and continuous action environments (TRPO). For all environments trained with Double-DQN (*Tetris*, *VizDoom*, *HauntedHouse*) we use a fixed episode length of $500$ for training and collect $1000$ sample between training rounds that perform $1000$ gradient steps on the network. The replay buffer size that is used is $50000$. The same size is used for additional data buffers for RND and ICM. For *Tetris* and *HauntedHouse* network with layer sizes $[128, 64, 32]$ is used for both Q-networks. For *VizDoom* the network include 3 additional convolutional layers with $[64, 32, 8]$ filters with strides $[5, 4, 3]$, all using relu activations. A learning rate of $0.003$ is used to train the Q networks.

For the *Humanoid* environments the network uses relu activations with hidden layer sizes $[256, 128]$. TRPO is used to train the policy with the advantage estimated with Generalize Advantage Estimation. The training collects $4098$ sample at a time, performs $64$ gradient steps on the value function and one step with TRPO. A fixed variance is used for the policy of $0.2$ which is scaled according to the action dimensions from the environment. Each episode consisted of $4$ rounds of training and it typically take $20$ hours to train one of the SMiRL policies using $8$ threads. A kl constraint of $0.2$ is used for TRPO and a learning rate of $0.001$ is used for training the value function. Next, we provide additional details on the state and action spaces of the environments and how $\theta$ was represented for each environment.

**Tetris**    We consider a $4 \times 10$ *Tetris* board with tromino shapes (composed of 3 squares). The observation is a binary image of the current board with one pixel per square, as well as an indicator integer for the shape that will appear next. A Bernoulli distribution is used to represent the sufficient statistics $\theta$ given the to policy for SMiRL. This distribution models the probability density of a block being in each of the boad locations. Double-DQN is used to train the policy for this environment. The reward function used for this environment is based on the Tetris game which gives more points for eliminating more rows at a single time.

**VizDoom**    For the *VizDoom* environment the images are scaled down to be $48 \times 64$ grayscale. Then a history of the latest $4$ images are stacked together to use as in separate channels. To greatly reduce the number of parameters $\theta$, SMiRL needs to estimate in order to compute the state entropy the image is further reduces to 20. A Gaussian distribution is used to model the mean and variance over this state input. This same design is used for *TakeCover* and *DefendTheLine*. An episode timelimit of $500$ is used for each environent. Double-DQN is used to train the policy for this environment.

**Simulated *Humanoid* robots.** A simulated planar *Humanoid* agent must avoid falling in the face of external disturbances (Berseth et al., 2018). The state-space comprises the rotation of each joint and the linear velocity of each link. We evaluate four versions of this task: *Cliff*, *Treadmill*, *Pedestal*, and *Walk*. The *Cliff* task initializes the agent at the edge of a cliff, in a random pose and with a forward velocity of 1 m/s. Falling off the cliff leads to highly irregular and unpredictable configurations, so a surprise minimizing agent will want to learn to stay on the cliff. In *Treadmill*, the agent starts on a platform that is moving backwards at 1 m/s. In *Pedestal*, random forces are applied to it, and objects are thrown at it. In this environment, the agent starts on a thin pedestal and random forces are applied to the robot's links and boxes of random size are thrown at the agent. In *Walk*, we evaluate how the SMiRL reward stabilizes an agent that is learning to walk. In all four tasks, we evaluate the proportion of episodes the robot does not fall. A state is classified a fall if the agent's links, except for the feet, touch the ground, or if the agent is $-5$ meters or more below the platform or cliff. Since the state is continuous, We model $p_\theta(\mathbf{s})$ as independent Gaussian for these tasks. The full pose and link velocity state is used for the *Humanoid* environments $\theta$. The simulated robot has a control frequency

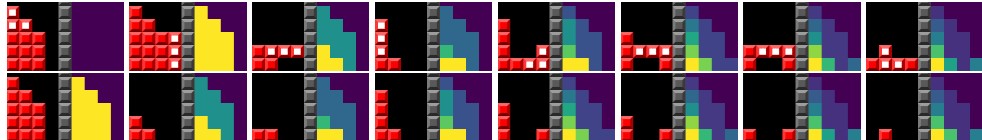

Figure 8: Frames from *Tetris*, with state **s** on the left and parameters $\theta_t$ of an independent Bernoulli distribution for each board location on the right, with higher probability shown in yellow. The top row indicates the newly added block and bottom row shows how the state changes due to the newly added block along with the updated $\theta_t$.

of $30\text{hz}$. TRPO is used to train the policy for this environment. Similar to *VizDoom* $p(\mathbf{s})$ is modeled as an independent Gaussian distribution for each dimension in the observation. Then, the SMiRL reward can be computed as:

$$r_{\text{SMiRL}}(\mathbf{s}) = -\sum_i \left( \log \sigma_i + \frac{(\mathbf{s}_i - \mu_i)^2}{2\sigma_i^2} \right),$$

where **s** is a single state, $\mu_i$ and $\sigma_i$ are calculated as the sample mean and standard deviation from $\mathcal{D}_t$ and $\mathbf{s}_i$ is the $i^{th}$ observation feature of **s**.

***HauntedHouse*.** This partially observed navigation environment is based on the *gym_minigrid* toolkit (Chevalier-Boisvert et al., 2018). The agent vision if changed to be centered around the agent. The experiments in the paper combine SMiRL with curiosity measures for *Counts* that are computed using the agent locations in the discrete environment. Similar, to the *VizDoom* and *Humanoid* environments a Gaussian distribution over the agents observations is used to estimate $\theta$. Double-DQN is used to train the policy for this environment.

**SMiRL VAE training** The encoders and decoders of the VAEs used for *VizDoom* and *Humanoid* experiments are implemented as fully connected networks. The coefficient for the KL-divergence term in the VAE loss was $0.1$ and $1.0$ for the *VizDoom* and *Humanoid* experiments, respectively. We found it very helpful to train the VAE in batches. For the *Humanoid* experiments where TRPO is used to train the policy the VAE is trained every 4 data collection phases for TRPO. This helped make the learning process more stationary, increasing convergence. The design of the networks used for the VAE mirrors the size and shapes of the policies used for training described earlier in this section.

**Fixed Length Episodes** For SMiRL it helped to used fixed length episodes during training to help keep SMiRL from terminating early. For example, in the *VizDoom* environments SMiRL would result in policies that would terminate as soon as possible so the agent would return to a similar initial state. In fact, for training we need to turn on god mode to prevent this behaviour. Similarly, to discourage SMiRL from terminating Tetris early by quickly stacking pieces in the same tower (resulting in low entropy) we added "soft resets" where the simulation will reset when the game fails and the episode will continue on forcing the SMiRL agent to learn how to eliminate rows to reduce the number of blocks in the scene.

## C  SMiRL Distributions

**SMiRL on *Tetris*.** In *Tetris*, since the state is a binary image, we model $p(\mathbf{s})$ as a product of independent Bernoulli distributions for each board location. The SMiRL reward $\log p_\theta(\mathbf{s})$ becomes:

$$r_{\text{SMiRL}}(\mathbf{s}) = \sum_i \mathbf{s}_i \log \theta_i + (1 - \mathbf{s}_i) \log(1 - \theta_i),$$

where **s** is a single state, the update procedure $\theta_i = \mathcal{U}(\mathcal{D}_t)$ returns the sample mean of $\mathcal{D}_t$, indicating the proportion of datapoints where location $i$ has been occupied by a block, and $s_i$ is a binary variable indicating the presence of a block at location $i$. If the blocks stack to the top, the game board resets, but the episode continues and the dataset $\mathcal{D}_t$ continues to accumulate states.

| Environment | RND | Random | SMiRL | Relative |
|---|---|---|---|---|
| *Tetris* | 18.6±2.7 | 17.1±1.8 | 5.2±2.1 | -43.4 |
| *TakeCover* | -4.7±0.7 | -5.9±1.1 | -13.2±0.7 | -10.4 |
| *DefendTheLine* | 19.6±0.6 | 19.9±0.7 | -23.4±0.4 | -8.5 |
| *Assault* | 193.1±1.4 | 181.8±2.7 | 124.9±2.3 | -45.6 |
| *SpaceInvaders* | 208.4±3.4 | 206.5±5.2 | 196.3±4.2 | -8.3 |
| *Carnival* | 151.2±1.4 | 130.8±2.7 | 107.7±4.3 | -2.7 |
| *RiverRaid* | 264.4±3.4 | 269.1±2.2 | 274.9±3.2 | 0.3 |
| *Gravitar* | 198.6±1.7 | 167.8±2.7 | 141.3±1.3 | 4.3 |
| *Berzerk* | 197.2±1.4 | 180.0±2.7 | 177.1±4.7 | 14.3 |

Table 2: Estimated entropies for three of our tasks, and an example Atari games studied by Burda et al. (2018b), where novelty-seeking exploration works well. Note the large *negative Relative* entropy gap in our tasks with overall lower initial entropy, which are both absent in most Atari games. This data shows the mean and std over 3 seeds.

## D   SMiRL MDP

Note that the RL algorithm in SMiRL is provided with a standard stationary MDP (except in the VAE setting, more on that below), where the state is augmented with the parameters of the belief over states $\theta$ and the timestep $t$. We emphasize that this MDP is Markovian, and therefore it is reasonable to expect any convergent reinforcement learning (RL) algorithm to converge to a near-optimal solution. Consider the augmented state transition $p(s_{t+1}, \theta_{t+1}, t+1 | s_t, a_t, \theta_t, t)$. This transition model does not change over time because the updates to $\theta$ are deterministic when given $s_t$ and $t$. The reward function $r(s_t, \theta_t, t)$ is also stationary, and is in fact deterministic given $s_t$ and $\theta_t$. Because SMiRL uses RL in an MDP, we benefit from the same convergence properties as other RL methods.

**Transition dynamics of $\theta_t$.**   Given the augmented state $(\mathbf{s}_t, \theta_t, t)$, we show that the transition dynamics of the MDP are Markovian. The $\mathbf{s}_t$ portion of the augmented state are from the environment, therefore all convergence properties of RL hold. Here we show that $(\theta_t, t)$ is also Markovian given $\mathbf{s}_{t+1}$. To this end, we describe the transition dynamics of $(\theta_t, t)$ for an incremental estimation of a Gaussian distribution, which is used in most experiments. Here we outline $\theta_{t+1} = \mathcal{U}(\mathbf{s}_t, \theta_t, t)$.

$$\theta_t = (\mu_t, \sigma_t^2)$$
$$\mu_{t+1} = \frac{t\mu_t + \mathbf{s}_t}{t+1}$$
$$\sigma_{t+1}^2 = \frac{t(\sigma_t^2 + \mu_t^2) + \mathbf{s}_t}{t+1} - \mu_{t+1}^2$$
$$\theta_{t+1} = (\mu_{t+1}, \sigma_{t+1}^2)$$
$$t_{t+1} = t_t + 1$$

These dynamics are dependant on the current augmented state $(\mathbf{s}_t, \theta_t, t)$ and the next state $\mathbf{s}_{t+1}$ of the RL environment and do not require an independent model fitting process.

However, the version of SMiRL that uses a representation learned from a VAE is not Markovian due to not adding the VAE parameters to the state $\mathbf{s}$, and thus the reward function changes over time. We find that this does not hurt results, and note that many intrinsic reward methods such as ICM and RND also lack stationary reward functions.

## E   MORE ENVIRONMENT STABILITY DETAILS

Here we include the full data on the stability analysis in Figure 2. From this data and the additional results on the website we can see the SMiRL can reduce the entropy of a few of the Atari environments as well. These include Assault, where SMiRL hides on the left but is good at shooting ships and Carnival, where SMiRL also reduces the number of moving objects. RND on the other hand tends to induce entropy and cause many game flashes.

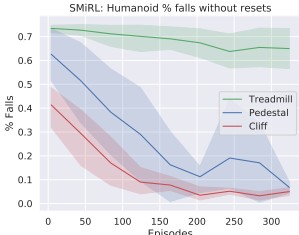

Figure 9: SMiRL results when the *Humanoid* environments are trained without early termination based resets (fixed episode lengths). *Cliff* and *Pedestal* still produce entropy minimizing policies that reduce falls. RL has difficulty with optimizing the more challenging *Treadmill* environment.

## F    ADDITION NOTES ON UNSUPERVISED RL RELATED WORK

The works in Tschantz et al. (2020b); Annabi et al. (2020) are interesting and discuss connections to active inference and RL. However, these methods and many based on active inference "encode" the task reward function as a "global prior" and minimizing a KL between the agents state distribution this "global prior". Our work instead actively estimates a marginal over the distribution of states the agent visits (with no prior data) and then minimizes this "online" estimate of the marginal, as is described in Section 3. Our work differs from LP-based methods (Kim et al., 2020; Lopes et al., 2012; Schmidhuber, 1991) because SMiRL is learning to control the marginal state distribution rather than identifying the system parameters.

## G    ADDITIONAL RESULTS

To better understand the types of behaviors SMiRL produces we conducted an experiment with fixed episode lengths on the *Humanoid* environments (Figure 9). This shows that SMiRL results in surprise minimizing behaviors independent of how long the episode is.

