# OpenReview forum: "SMiRL: Surprise Minimizing Reinforcement Learning in Unstable Environments"
_ICLR.cc/2021/Conference — ICLR 2021 Oral_

### Official Review · AnonReviewer4 · 2020-10-20
**Interesting approach, but several issues need to be addressed**

**Rating:** 7
**Confidence:** 4

**Review:**

This paper presents and studies an unsupervised learning approach to the emergence of "meaningful/useful"
behaviour in a Deep RL setting, based on an algorithm which objective consists in minimizing surprise.
It is possible to see this paper as studying a Deep RL implementation of cognitive homeostasis, an old idea
recently popularized through the work on the free energy principle by Friston and colleagues.
The paper presents two versions of this algorithm, one without representation learning, and one with
representation learning (VAE), and compares it to Deep RL algorithms that maximize prediction error/novelty
from two perspectives: emergence of behaviours and utility as an auxiliary reward to solve sparse reward problems.
The paper discusses the relevance of the approach in environments that are said to be "unstable" (with an attempt
to formalize this concept). The paper also shortly presents an application of the algorithm for imitation learning.
Experiments are presented in a variety of environments.

Major strengths:
+ The paper is globally clear and well-written
+ The qualitative results showing and analyzing emergent behaviour are stimulating
+ Implementation of the free-energy principle in high-dimensional spaces is known to be challenging and this
paper contributes towards understanding how to do it (yet the precise formal articulation between SMIRL and the
FEP remains to be done, and this does not seem to be the primary objective of the paper)
+ The discussion of the complementarity of within-episode SMIRL and across-episode novelty seeking intrinsic
rewards is interesting
+ The shortly described application for imitation learning is promising
+ The are many and varied environments in experiments

Major weaknesses:
- The environments chosen for experimentation are such that minimizing surprise aligns very well with either
producing "interesting" behaviour or maximizing an external reward. One could easily consider slight modifications
of most environments where a trivial solution to minimizing surprise could be found instead (e.g. giving the ability to push
on the "pause" button in the Tetris game or having safe rooms in opposite directions than enemies in Vizdoom/HauntedHouse).
In a real robot, this would lead a robot to hide in the corner of a room and just look at a uniform wall that does not move.
This problem of trivial solutions to surprise minimization approaches has been called the Dark Room problem in
the FEP literature. This is known to be a limit of this general approach when applied to the real world, and how to address it
operationnally (in a tractable manner) under this paradigm remains an open question.
- Related to the point above, the fact that different environments are used in different experiments of the paper does not
allow to get a good grasp of the general behaviour of the algorithm: I would recommend to run all experiments in all
environments (maybe to report in Annex), and at least to justify why environmnets change across experiments.
-  The paper is positioned in comparison with novelty seeking/prediction error maximizing agents, and motivates
the surprise minimizing approach by arguing that real-world situations are problematic for novelty seeking agents,
as the real world spontenously generates novelty through distractors that should be avoided rather than explored.
This is accurate, but an incomplete discussion of the literature in the area of unsupervised learning of behaviours:
approaches maximizing learning progress were precisely introduced to adress these limits of novelty seeking approaches
and to scale to real world environments with two families of distractors: novel unlearnable parts of the environments
(which is a problem for novelty seeking approaches) and trivial low-entropy parts of the environments (which are a problem
in principle for surprise minimization approaches). Thus, the paper should position itself in comparison with LP-based
approaches and show experiments with at least some of the existing LP-based algorithms (e.g. Schmidhuber, 1991; Lopes et al., 2012;
Kim et al., 2020).
- As the authors acknowledge, the version of SMIRL using represention learning is not fully "within episode surprise minmization",
as the learned representation depends on past episodes: as a consequence, it is unclear what one should conclude about the relation
between within vs across episode mechanisms and the properties of emergent behaviours
- The paper lacks discussion of related work aiming to implement surprise minimization approaches that scale to high-dimensional
spaces, especially in a RL framework, e.g. Tschantz et al., 2020; Annabi et al., 2020
- The paper does not provide code to enable reproducibility of results (and does not say it will)

Overall, the topic of this paper is interesting and the work could make a valuabe contribution, especially from the perspective
of studying incentives to emergent structured behaviour in unsupervised learning,. However, the weaknesses mentionned above need
to be addressed to better establish the contributions of this paper, both in terms of understanding the generality of the
results and the contributions in relation to the existing litterature.

References:

Annabi, L., Pitti, A., & Quoy, M. (2020). Autonomous learning and chaining of motor primitives using the Free Energy Principle. arXiv preprint arXiv:2005.05151.
Kim, K. H., Sano, M., De Freitas, J., Haber, N., & Yamins, D. (2020, January). Active world model learning in agent-rich environments with progress curiosity. In International Conference on Machine Learning (ICML).
Lopes, M., Lang, T., Toussaint, M., & Oudeyer, P. Y. (2012). Exploration in model-based reinforcement learning by empirically estimating learning progress. In Advances in neural information processing systems (pp. 206-214).
J. Schmidhuber, “Curious model-building control systems,” in Proc. Int. Joint Conf. Neural Netw., Singapore, 1991, vol. 2, pp. 1458–1463.
Tschantz, A., Baltieri, M., Seth, A. K., & Buckley, C. L. (2020, July). Scaling active inference. In 2020 International Joint Conference on Neural Networks (IJCNN) (pp. 1-8). IEEE.
Tschantz, A., Millidge, B., Seth, A. K., & Buckley, C. L. (2020). Reinforcement Learning through Active Inference. arXiv preprint arXiv:2002.12636.

---

> ### Author Response · Authors · 2020-11-16
> **Responce for R4**
>
> Reviewer 4 Rebuttal:
> We thank the reviewer for their time and comments on the paper. We’ve modified the paper to address the following issues raised in the review. (1) The related work section has been updated to include the suggested citations and a discussion of the Dark Room issue with FEP based methods. We explain these changes below, with a more detailed discussion in the appendix (due to space constraints). (2) In regard to experiments and environments, we are currently running additional experiments to include all domains for each section and will add them to the paper. (3) With respect to missing discussion and comparison to learning progress (LP) approaches, we have included discussion about their differences, and are in the process of adding additional experiments that use an LP-based method. (4) Lastly, we have released the code for the paper on the website (https://sites.google.com/view/surpriseminimization) and included more details in Appendix A on the training algorithms, networks and hyperparameters. We believe that these additions and modifications address all of the concerns raised in the review, but please let us know if there are any further issues to address.
>
> ----------------------------------------
> Q1: Dark Room problem in the FEP literature (trivial entropy minimizing solutions)
>
> A1:The Dark Room problem is a known issue for Free Energy Principle (FEP) based methods and surprise minimization methods. As discussed in Table 1, we do not aim for our method to work in either partially-observed environments or environments that do not satisfy our unstable assumption. In practice, we expect the number of truly dark rooms in the real world to be low, as the robot’s internal state (battery life, motor drift) should always be observable, but we agree that this is an important potential issue.  We have added content to the Related Work section to note this issue and updated Section 4 to further describe the types of environments in which SMiRL should work well.
>
> ----------------------------------------
> Q2: Different environments are used in different experiments or  justify why environments change across experiments
>
> A2: We will run the missing experiments and add them to the paper. The missing experiments are the VAE-based experiments in the Tetris environment and SMiRL + ICM on Tetris and Vizdoom. We do not expect that the VAE will help for Tetris because the Tetris observation is already well abstracted. For both Tetris and Vizdoom, we did not find that SMiRL alone suffered from a lack of exploration, but it is possible that adding ICM could speed up learning.
>
> ----------------------------------------
> Q3: Discussion of the literature in the area of unsupervised learning of behaviours
> A3: We thank the reviewer for the suggestion of positioning our work in relation to LP-based methods, which direct the agent to investigate complex but learnable dynamics in the environment. We have revised the related work section and the appendix to include a discussion of these methods. Our work differs from LP-based methods because SMiRL is learning to control the marginal state distribution rather than identifying the system parameters. We are investigating running experiments for an LP-based method, however, the papers cited do not include code. Would using the “VIME: Variational Information Maximizing Exploration” method be acceptable in this regard?
>
> Q4: Discussion of active-inference based surprise minimization in high-dimensional spaces, e.g. Tschantz et al., 2020; Annabi et al., 2020
> A4: We added text on the works by Tschantz et al. 2020 and Annabi et al. 2020 in the related work. These approaches require the task reward function as a “global prior” and minimize a KL between the agent’s observation distribution and this “global prior”. Our work is instead in the area of unsupervised reinforcement learning (i.e. in the absence of task rewards).

---

> > ### Comment · AnonReviewer4 · 2020-11-18
> > **Follow-up discussion**
> >
> > Hi,
> >
> > thanks a lot for all the answers.
> >
> > > Q1: Dark Room problem in the FEP literature (trivial entropy minimizing solutions)
> >
> > I think it is indeed much better that the paper now discusses this issue. I disagree with authors about the fact that the real world should contain few dark rooms, as even if a robot has access to is energy level it may find some strategies to remain pluggued (or go back to plug easily), close the eyes, and do nothing else. However, as the issue is acknowledged in the paper, I do not think this is a problem here: it is a more general debate outside the scope of this work.
> >
> > > we have released the code
> >
> > Great! It is not very well documented at this point, I encourage you to do so at a later point.
> >
> > >Q2: Different environments are used in different experiments or justify why environments change across experiments
> > > We will run the missing experiments and add them to the paper.
> >
> > Thanks, I am looking forward to see the full set of experiments.
> >
> > > Q3: Discussion of the literature in the area of unsupervised learning of behaviours A3: We thank the reviewer for the suggestion of positioning our work in relation to LP-based methods
> >
> > I appreciate the answer, contrasting the objective of controlling the marginal state distribution rather than identifying the system parameters.
> > From this perspective, it seems that both novelty-based and LP-based intrinsic motivation approaches can be seen as complementary to the within-episode surprise minimization intrinsic motivation mechanism you propose. Like R3, I think explaining this would be very useful in the introduction. I will be interested by experiments using LP-based approaches, but given the code of the paper I mentioned does not seem to be available, and given the positioning in terms of complementarity (and control of marginal state distrib. vs identifying systems parameters), it seems ok if you're not including it.
> >
> > Furthermore, I do not think authors responded to my question about the within- versus across- episode issue with the version of SMIRL using representation learning.
> >
> > Also, it would be useful to discuss in more details links with empowerment intrinsic motivation: re-reading the paper, it seems to me that another approach in the litterature that has strong links to SMIRL is empowerment-based intrinsic motivation, which has links with suprise minimization and FEP (see for e.g. the short discussion in Biehl et al., 2015), in particular for the within-epidose perspective.
> >
> > References:
> >
> > Biehl, M., Guckelsberger, C., Salge, C., Smith, C., & Polani, D. (2015). Free energy, empowerment, and predictive information compared. Technical report, University of Hertfordshire. URL: https://www. mis. mpg. de/fileadmin/pdf/abstract gso18 3300. pdf.

---

> > > ### Author Response · Authors · 2020-11-20
> > > **R4 Discussion**
> > >
> > > Q5: within- versus across- episode SMiRL with representation learning
> > >
> > > A5 (updated): Our study is in the setting in which a reliable state representation is available or easily learned from existing observation. Training the learned state representation online does affect the approximation in Eq (1) as the learning process is changing the representation for which the entropy is computed. While we do not have any theoretical analysis of how this representation learning process interacts with our algorithm, empirically we did not find this non-stationarity to be an issue, and generally observed learned representation to improve performance in experiments with large image observations or complex dynamics. Using these learned representations is not meant to address the difficulty of estimating and minimizing surprise across episodes or more generally over long sequences (possibly a single episode) which is a challenge for surprise minimization-based methods. We are happy to comment on this more if there are additional questions. We have expanded our existing discussion on this limitation to clarify this point.
> > >
> > >
> > > Additional related work:
> > > Thank you, the work in Biehl et al., 2015 provides a discussion on connecting FEP, empowerment and Predictive information maximization in the same framework to explain their similarities and differences. We have added this content to the related work section.

---

> > > ### Author Response · Authors · 2020-11-23
> > > **RE: Additional Experiments**
> > >
> > > We have completed the requested additional experiments. See the additional analysis in Figure 3 and three new plots in Figure 3. We have also updated the paper text to integrate these results. With these additional experiments, the conclusions remain largely the same. Here we provide a summary of these additional experiments.
> > > - Emergent Behavior:
> > >   - Added SMiRL + VAE on Tetris (Figure 3 top-left and top-center) and DefendTheLine (Figure 3 bottom-left)
> > >   - Added SMiRL + ICM on Tetris (Figure 3 top-left and top-center) and TakeCover (Figure 3 top-right ) DefendeTheLine (Figure 3
> > >   - Evaluation on DefendTheLine (Figure 3 bottom-left)
> > > bottom-left)
> > > - Auxiliary reward experiments
> > >   - Performed this for the Tetris environment (Figure 3 bottom-center)
> > >   - Performed this for the VizDoom TakeCover environment (Figure 3 bottom-right)

---

> > > > ### Comment · AnonReviewer4 · 2020-11-24
> > > > **Follow-up discussion**
> > > >
> > > > Thanks for all the updates! I think overall they have significantly improved the paper, which I think now explains well its contributions and will be of interest to many readers. I raised my score accordingly.

---

### Official Review · AnonReviewer3 · 2020-10-23
**This paper introduces suprise minimization as a intrinsic objective. The idea is novel, well motivated and seem to be supported by empirical evidence. However, some details are missing and there might be a confounding factor.**

**Rating:** 7
**Confidence:** 4

**Review:**

------------------------------------
**Summary:**

This paper proposes a new intrinsic objective for RL agents: surprise minimization. This may come as a surprise, as other related works usually propose to maximize surprise, or to maximize novelty. The authors present motivations and conduct an empirical study on several environments to support their idea.

------------------------------------
**Strong points:**

Overall, I think it is a good paper. Let me list some strong points:

* The idea is simple, novel and well motivated. The paper positions this new intrinsic objective with respect to variants of novelty maximization objectives and brings a new perspective.
* It’s well written and organized
* The algorithm is tested on a relevant selection of environments and against state-of-the-art algorithms using intrinsic objectives.
* The empirical evidence seems to support the claims.
* The related work is quite complete.
* I liked the discussion about stable vs unstable environments. It is the first time I see it discussed.
* The website brings visualizations of trained policies.
------------------------------------
**Weak points:**

I will now list a few weak points of the paper.
* Some descriptions of the results are missing. In figures, what does the shaded area represent? (std, sem, confidence intervals, etc). In Table 1, what are the numbers? (what is the central tendency, what is the error, how many seeds?). Same for Appendix D.
* The paper does not provide all necessary information to reproduce the results. There is no detail about the RL algorithms (TRPO and DQN), no description of the architectures, and no hyperparameters. This is important and should be contained somewhere in the Appendix.
* Will the code be released? If not, why so? Same questions for the environments, are they accessible somewhere?
* It seems to me that this approach could potentially tackle harder problems, but the paper is limited to a simplification of the tetris game, planar humanoid variants and Doom. The x-axes of the figures also tell us that only a few episodes were needed to solve them. I am not saying that I need the hardest games solved to find an algorithm interesting, but I am wondering whether it would scale. Could you tell us whether you attempted to tackle harder environments, and if so, why do you think SMiRL failed? I think we can gain a lot of understanding by looking at negative results.
For example, I feel like testing this algorithm on a 3D humanoid would better demonstrate the power of this approach.
* It seems to me that there might be a confounding factor that could partially explain the success of the surprise reward. Indeed, it seems that all environments presented here can terminate when the agent dies. This is a guess, as I could not find this information in the paper (please add it). If so, then the expected cumulative rewards is an increasing function of the lifetime of the agent in the game. Because of this, maximizing the cumulative surprise might be a good idea because it goes in the same direction (by construction) as maximizing survival.
A counter-argument from the paper can be the performance of SMiRL + reward in Walk, that is superior to the performance of reward alone. However, this is unclear as I could not find the description of the reward function in the paper (please add it).
Another way to disprove this hypothesis would be to compare the performance of SMiRL to the performance of an agent maximizing a reward function that gives +1 whenever the agent is alive (survival bonus). If it performs better, then surprise maximization brings something to the table, if it does not, then it might work because it is correlated to the survival time.
Another way could be to have episodes that do not include death-related resets (fixed length episodes).

------------------------------------
**Recommendation and justification:**
This is overall a strong paper. However I’m concerned about the potential confounding factor of the survival time. I’ll give a score of 6, but I would happily increase that score if
1. The authors convince me that the success of the surprise maximization is not due to the survival confounding factor.
2. The authors include all necessary details for reproducibility and/or release the code.
3. Discuss the scalability to harder problems (e.g. 3D Humanoid).

------------------------------------
**Feedback to improve the paper (not part of assessment)**

* I would move the discussion about how surprise minimization and novelty maximization can be complementary to the intro. These two approaches seem to go in opposite directions and, as a reader, I would be happy to read this discussion early.
* It would be interesting to discuss how it plays out in natural agents. The intuition is that minimizing surprise leads to finding a stable configuration and staying there. In practice, it is probably balanced with other driving needs like the need for food. I guess it is discussed in related papers like Friston 2009. In natural agents, surprise minimization must also be model-based. Indeed, animals do not need to jump out of cliffs several times to know that it’s bad.
* Not sure I understand the inequality in Eq1. Maybe I missed something.
* Discuss the surprise maximization approach of Achiam et al and whether it differs from yours.
* Table 1: the legend seems to disagree with the results. It seems to me that the entropy difference is as low in your environment as in the others, but the caption says “note the clear negative  entropy gap on our tasks, whereas this clear trend is absent on the Atari games”.
* Fig 4. Is it really training in 80 episodes? There are very few images to train a VAE, especially if the episode resets when the agent falls (does it?) How many steps per episode?
* How do you get demos for humanoid tasks? I guess there are not human demos but previously trained agents?
* Results in Fig. 6 are not super satisfying, they are quite far from the target (although I guess it is a difficult task). I am not even sure the second example is achievable. In the traditional Tetris, wouldn’t cubes fall due to gravity?

**Typos:**
* “our results are available online” → missing full stop.
* “In such environments, which we believe are more reflective of the real world” → previous sentences do not discuss environments but intrinsic objectives.
* “unexpected events don’t happen” → “do not”
* “deep DQN” → “Deep Q-Networks”, or “DQN”


**Update post-rebuttal**
The authors addressed most of my concerns, especially the one about the confounding factor. I am updating the score from 6 to 7.

---

> ### Author Response · Authors · 2020-11-16
> **Responce to R3**
>
> Thank you very much for your time and valuable feedback!
>
> Q1: Is SMiRL effective only by providing a survival bonus to avoid death-related resets?
>
> A1: In our current experiments, we use fixed-length episodes during training for all Tetris, VizDoom and HauntedHouse experiments. The use of fixed-length episodes avoids the issue of SMiRL appearing to be a survival bonus, as there are no death-related resets in those experiments. The humanoid experiments do use death-related resets. However, for these environments, a survival bonus would treat the desired behaviour of remaining on the cliff, treadmill, or pedestal the same as falling off. We have added these details on the use of fixed-length episode training to Appendix A.
>
> ----------------------------------
> Q2: Information to reproduce the results.
> A2: We have addressed this in two ways. First, additional information has been added on  environments, training networks and hyperparameters in the appendix. Second, we have posted the code on the paper website (https://sites.google.com/view/surpriseminimization) and will include it with the published paper.
>
> -------------------------------
>
> Q3: Harder problems for SMiRL
> A3: We agree that running SMiRL in more interesting and difficult environments could provide additional information. We have not investigated many environments beyond the ones used in the paper. However, as per your request, we are running SMiRL on a 3D humanoid and will include these results in the paper as well.
>
> ----------------------------------
> Q4: Clear entropy gap and how are the values computed.
>
> A4: We examined a set of environments, Tetris, Vizdoom and a collection of Atari games, to determine their relative entropy gap. We found that the environments we focused on in the paper (DefendTheLine, Tetris, TakeCover) do have negative relative entropy gaps, shown in Table 1. In addition, we found that two Atari environments (Assult and SpaceInvaders) also have a negative entropy gap, indicating that SMiRL should work well on them. We have revised the wording for the caption to clarify this relationship.
> The measure used to approximate the entropy is the same as the right side of equation (1) multiplied by -1. We added this detail to the last paragraph of section 4 as well. In the table, we provide the mean and std over 3 random seeds for each method on each environment.
>
> -------------------------------
> Q5: What does the shaded area represent in the figures?
> A5: The shaded areas in the figures indicate the std of the random seeds for each experiment. We have added notes to the figures to explain this.
>
> -------------------------------
> Q6: Compare to  the surprise maximization approach of Achiam et al
> A6: Thank you for bringing this paper to our attention. This paper maximizes surprise to encourage exploration and increase entropy, whereas our method does the opposite, and minimizes surprise. The paper’s goal is similar to RND and ICM, in that it is aimed at promoting visitation of novel states, in this case by training a model over recent experience and adding a bonus to the reward when the model does not predict the data well (because the agent is in a novel area of the state space). We have included this in Section 2 of the paper to make our review of related work more thorough.
>
> -------------------------------
> Q7: Fig 4. Is it really training in 80 episodes?
> A7: There was a multiplication error, the number of episodes is actually 4 times that amount. We have updated the figures in the paper.
>
> -------------------------------
> Q8: How do you get demos for humanoid tasks?
> A7: We use the policy that was trained using the reward function (Reward in Figure 7) to collect 2048 samples of states to initialize \theta with. This is described briefly in “SMiRL as an auxiliary reward” section. We have provided more details on how this data is collected and used in Appendix A.
>
> -------------------------------
> Q9: “Results in Fig. 6 are not super satisfying, they are quite far from the target (although I guess it is a difficult task)”
> A9: SMiRL does fail to reach the targets in Fig 6, but please consider looking at the videos at https://sites.google.com/view/surpriseminimization, linked in the paper, which may provide a better sense of the performance on this task and the difficulty involved. Also note that for the second row of Fig 6, the target checkerboard pattern is actually impossible to achieve in the Tetris game.

---

> > ### Comment · AnonReviewer3 · 2020-11-16
> > **Answer from R3**
> >
> > Thank you for this detailed answer.
> >
> > **Q1: Is SMiRL effective only by providing a survival bonus to avoid death-related resets?**
> > Thank you for answering this concern. Using fixed-length episodes does correct for the survival bias I agree. However I'm not sure I understand the point in Humanoid tasks. It does use death-resets, but how do you define death here, if it is not falling of ?
> >
> > **Q2: reproducibility concerns**
> > Great.
> >
> > **Q3: more difficult environments**
> > Thank you for running these experiments. I'm looking forward to the results.
> >
> >
> > **Others**
> > Thank you for the other answers and modifications.

---

> > > ### Author Response · Authors · 2020-11-16
> > > **Follow up Q1**
> > >
> > > Thank you for your quick response.
> > >
> > > A1: For the humanoid tasks death occurs when the agent falls 5 meters below the ground. At that point, it is impossible for the agent to recover and balance on the ground. These resets prevent the agent from using time, collecting data in this bad area of the state space and as a result speed up learning. However, we are running experiments without these resets and will include the results in the paper.

---

> > > ### Author Response · Authors · 2020-11-23
> > > **Followup**
> > >
> > > Re: SMiRL vs survival reward.
> > >
> > > After rerunning the humanoid experiments without resets, most environments still learn to minimize entropy with fixed episode length. See Figure 9 at the end of the paper.
> > >
> > > Re: 3D biped results.
> > >
> > > We have attempted to produce 3D biped results. Thus far, we have not been able to produce policies that significantly reduce entropy. This may be due to the increased complexity of the environment and much larger action space.

---

### Official Review · AnonReviewer1 · 2020-10-27
**The authors target the unsupervised reinforcement learning problem. An opposite idea from the existing approaches by maximizing state entropy is adopted to minimize state entropy. It is interesting that such an idea has achieved good performance in unstable environments.**

**Rating:** 8
**Confidence:** 4

**Review:**

The authors target the unsupervised reinforcement learning problem. An opposite idea from the existing approaches by maximizing state entropy is adopted to minimize state entropy. It is interesting that such an idea has achieved good performance in unstable environments. A state distribution is fitted during the interaction with an environment and the probability of the current state is used as a virtual reward. The parameters or sufficient statistics are also applied to the policy. The motivation is clear and verified. It is generally a good paper.

It is surprising that the exploration is achieved in the long term even minimizing state entropy. Is that possible the exploration events are from the 'unstable' environment? What if there are some patterns underlying the exploration events but only part of the 'unstable' environment? Is that OK to totally rely on unexpected events from the environment to explore the environment? Is that possible to add some exploration strategy in the developed model?

---

> ### Author Response · Authors · 2020-11-16
> **Responce to R1**
>
> Thank you very much for the feedback!
>
> It is true the unstable environment does produce some underlying exploration but the algorithm does not rely on this noise. This may be verified by observing that all algorithms experience the unstable environment, yet SMiRL performs significantly better than the baselines across our experiments. As far as adding explicit exploration strategies to SMiRL, yes, this is possible, and we have experimented with this idea by adding traditional novelty-seeking bonuses to SMiRL (SMiRL + ICM) agents and obtained improved results in some environments (described in Pg 7, results in Figs 4 and 5).

---

### Official Review · AnonReviewer2 · 2020-10-28

**Rating:** 7
**Confidence:** 4

**Review:**

This work proposes an RL approach SMiRL that is able to learn effective policies in unstable environments without the need for external reward. The idea at a high-level is almost the opposite of intrinsic motivation RL approaches, which encourage novelty-seeking behaviors. The proposed method instead aims to minimize surprise or state entropy. To train the agent, rewards come from state marginal estimates, but because this distribution is changing, the authors create an augmented MDP. Through experiments on game domains and robot control tasks, the authors show that SMiRL outperforms intrinsic motivation methods. The authors also show that SMiRL can be used to do imitation and can be combined with regular reward signals.

Pros:
- The problem formulation is interesting and novel. Intrinsic motivation is well studied, but this problem considers the setting where the environment is unstable rather than static, which requires new methods.
- The paper is written well and is clear. The motivation is described well.
- The authors evaluate on many domains, highlighting the diversity of settings in which the approach can be applied.

Cons:
- It seems like this approach is only applicable to unstable environments. Does the approach fail for regular, static environments? I’m assuming the agent might just end up staying still because it’s trying to seek stable states. It can be combined with the external reward signal but will the minimizing entropy objective hurt you?
- Without common sense knowledge, this approach would take many iterations to learn. So while this formulation might be more similar to the real world, the real world would only allow for a few interactions with the world.

Comments:
- How is SMiRL doing better than the oracle in Figure 3 center?
- In Figure 3 left, it might be a problem that with minimal episodes, SMiRL does worse. If SMiRL is useful for more real-world unstable environments, this would require a simulator good enough to model the real world.

Recommendation:
Overall, the paper is interesting and novel. The approach is reasonable and experiments show the value of the method in unstable environments. I recommend “accept”.

------------
Post-rebuttal response:
The authors addressed most of my concerns so I continue to recommend acceptance of the paper. Specifically, they answered my question about whether the approach will work in static environments and how prior data can be used to improve sample efficiency. They also conducted additional experiments to verify some of my questions.

---

> ### Author Response · Authors · 2020-11-16
> **Clarifications and additions**
>
> Clarifications and additions
>
> Q1: SMiRL in static environments:
> A1: We acknowledge in Section 3, SMiRL is designed to work in unstable environments. In a completely static environment that does not evolve without agent action, a SMiRL agent would have no incentive to explore the environment, and may indeed avoid exploration under the influence of an external reward function, hurting performance. Note, however, that such misalignment between the intrinsic objective and an external reward signal is not specific to SMiRL in static environments; it would also occur for more traditional novelty-seeking methods in unstable environments. Section 4 has been updated to better describe this issue and the types of environments SMiRL should work well on.
>
> Q2: Sample-inefficient, requires realistic simulation and/or common sense:
>
> A2: While our method is indeed slower to learn on Tetris, this is not true across all environments. For environments that have sparse rewards (VizDoom Figure 7) or complex dynamics (humanoid, Figure 7 right) learning benefits from using SMiRL. However, we agree that SMiRL in its current form is sample-inefficient, and infusing prior knowledge is indeed one way to alleviate this issue. We show in Fig 7 that “Reward +SMIRL + prior data”, which has access to expert walking demonstration data for initializing the state distribution, learns a successful walking policy much faster than methods without access to such demonstration data. Section 5 has been edited to describe this and how using prior knowledge can be an avenue to speed up learning.
>
>
> -----------------------------
> Q3: “How is SMiRL doing better than the oracle in Figure 3 center?”
>
> A3: Figure 3 left and center show data from the same experiment that plots two different metrics, where the Oracle is optimized for minimizing deaths (left). The Oracle in our submission was trained with a reward that gives -1 when the player fails or dies (by stacking blocks to high) and 0 otherwise. In Figure 3 center, SMiRL performs better than the Oracle because the SMiRL objective, which wants to keep the board clear, aligns better with clearing rows than the Oracle’s goal of minimizing deaths. We have added additional information to Section 4 to explain the reward function to which the Oracle is optimizing. In Section 5, we have added more detail on how Figure 3 left and middle are the same experiment showing 2 different metrics.

---

> > ### Comment · AnonReviewer2 · 2020-11-18
> > **Answer from R2**
> >
> > Thanks to the authors for answering these questions!
> >
> > Q1 - Yes this makes sense. This was mainly to understand if the approach could be extended to handle static environments as well, but it's fine if the approach is best suited for unstable environments.
> >
> > Q2 - Thanks for expanding on this. Using demonstration data to speed up learning is a great step towards improving real-world sample efficiency.
> >
> > Q3 - If keeping the board clear is a better objective, why wouldn't the oracle be trained with this objective rather than one that minimizes deaths?

---

> > > ### Author Response · Authors · 2020-11-18
> > > **Q3 Oracle clarification**
> > >
> > > A3: Teris is not one of the common openAIGym or Atari benchmarks, however, the simple reward of -1 when the agent dies was seen as an easy and simple reward function to use to train the agent. We will run another experiment using rows cleared as the reward function and include that in Figure 3.

---

> > > > ### Comment · AnonReviewer2 · 2020-11-19
> > > > **Answer from R2**
> > > >
> > > > Thank you for your response - that would be great to see!

---

> > > > > ### Author Response · Authors · 2020-11-23
> > > > > **Additional Oracle Experiment**
> > > > >
> > > > > In Figure 3 (top-left and top-center), we included an additional experiment, Oracle (rows cleared), where the reward function used for the agent was the number of rows cleared. SMiRL also performs better than this Oracle.

---

### Decision · Program_Chairs · 2021-01-07
**Final Decision**

**Decision:**

Accept (Oral)

**Comment:**

The paper is studying a new intrinsic motivation RL setup in a dynamic environment, where the authors minimize the state entropy instead of the common approach of maximizing it. The resulting idea is simple but also surprising that it works so well. All reviewers appreciated the new problem formulation of using dynamic environments and found the idea very promsing. In addition, they identified the following strengths of the paper:
- The experiments are exhaustive, identifying many domains where the approach can be applied
- The presented results are compelling
- The paper is well written
- The paper introduces a new problem setup that has not been studied before

I agree with the reviewers that this paper contains many interesting contributions and therefore recommend acceptance.